# Exploring the Sensitivity of Subtropical Stand Aboveground Productivity to Local and Regional Climate Signals in South China

**Hua Zhou [1], Yang Luo [1], Guang Zhou [2], Jian Yu [2], Sher Shah [2], Shengwang Meng [3] and Qijing Liu [2,*]**

[1] Research Station of Ecology, Guizhou Academy of Forestry, No. 382 Fuyuan South Road Nanming District, Guiyang 550000, China; jeamourvous@163.com (H.Z.); luoyang0317@tom.com (Y.L.)
[2] Forestry College, Beijing Forestry University, No.35 Tsinghua East Road Haidian District, Beijing 100089, China; zhouguang910313@163.com (G.Z.); yujianyichen@outlook.com (J.Y.); shershah@bjfu.edu.cn (S.S.)
[3] Qianyanzhou Ecological Research Station, Institute of Geographic Sciences and Natural Resources Research, Chinese Academy of Sciences No. 11A, Datun Road, Chaoyang District, Beijing 100089, China; wangzai1220@126.com
[*] Correspondence: liuqijing@bjfu.edu.cn; Tel.: +86-139-1180-6586

**Abstract:** Subtropical forest productivity is significantly affected by both natural disturbances (local and regional climate changes) and anthropogenic activities (harvesting and planting). Monthly measures of forest aboveground productivity from natural forests (primary and secondary forests) and plantations (mixed and single-species forests) were developed to explore the sensitivity of subtropical mountain productivity to the fluctuating characteristics of climate change in South China, spanning the 35-year period from 1981 to 2015. Statistical analysis showed that climate regulation differed across different forest types. The monthly average maximum temperature, precipitation, and streamflow were positively correlated with primary and mixed-forest aboveground net primary productivity (ANPP) and its components: Wood productivity (WP) and canopy productivity (CP). However, the monthly average maximum temperature, precipitation, and streamflow were negatively correlated with secondary and single-species forest ANPP and its components. The number of dry days and minimum temperature were positively associated with secondary and single-species forest productivity, but inversely associated with primary and mixed forest productivity. The multivariate ENSO (EI Niño-Southern Oscillation) index (MEI), computed based on sea level pressure, surface temperature, surface air temperature, and cloudiness over the tropical Pacific Ocean, was significantly correlated with local monthly maximum and minimum temperatures (Tmax and Tmin), precipitation (PRE), streamflow (FLO), and the number of dry days (DD), as well as the monthly means of primary and mixed forest aboveground productivity. In particular, the mean maximum temperature increased by 2.5, 0.9, 6.5, and 0.9 °C, and the total forest aboveground productivity decreased by an average of 5.7%, 3.0%, 2.4%, and 7.8% in response to the increased extreme high temperatures and drought events during the 1986/1988, 1997/1998, 2006/2007, and 2009/2010 EI Niño periods, respectively. Subsequently, the total aboveground productivity values increased by an average of 1.1%, 3.0%, 0.3%, and 8.6% because of lagged effects after the wet La Niña periods. The main conclusions of this study demonstrated that the influence of local and regional climatic fluctuations on subtropical forest productivity significantly differed across different forests, and community position and plant diversity differences among different forest types may prevent the uniform response of subtropical mountain aboveground productivity to regional climate anomalies. Therefore, these findings may be useful for forecasting climate-induced variation in forest aboveground productivity as well as for selecting tree species for planting in reforestation practices.

**Keywords:** forests; mixed forests; climate change and disturbances; forest aboveground productivity; streamflow; MEI

## 1. Introduction

Due to the influences of the Tibetan-Himalayan Highland and Mediterranean climates, the subtropical zone in the Asian and European continents is mostly dry [1]. Only small coastal areas of the Western Pacific have sufficient moisture to ensure the development of subtropical forests, including South China and a chain of islands from Taiwan to Okinawa [1–3]. Therefore, the subtropical forests in South China are invaluable assets from a phytogeographical point of view [4]. A well-developed evergreen broadleaf forest exists in the eastern Nanling Mountains, which is the most extensive mountain range in South China. It is ecologically important to explore how the growth and aboveground productivity of the different forest stands change with the local-regional climate in the Nanling area. Climate change is a main important driver of short-term forest function [5] and long-term ecosystem dynamics [6], mainly as a result of expected warming [7]. The average temperatures increased by 1 °C globally during the past century because of human activities and global warming may reach 1.5 °C between 2030 and 2052 [8]. The influences of contemporary climate change on tree-ring growth, productivity, regeneration, and disturbance of subtropical forests have already been reported in several sites and ecosystems around the subtropical zone [2,3,9–14].

A broad range of subtropical forest types, including subtropical evergreen, deciduous broad-leaved, evergreen coniferous, and mixed deciduous forests, are distributed in the southern part of China. For historical reasons, natural forest resources in subtropical China have been seriously destroyed over the past few decades, and many primary forests have been changed to secondary forests or single-species forests [15,16]. Generally, establishing single-species stands is likely quicker and more economical, but there are many compelling reasons for establishing mixed-species forests. For example, most natural/primary forest stands are mixtures of two or more tree species [17,18]. Mixed-species stands can increase stress resilience and productivity under disturbances and extreme events better than single tree-species forests [19,20] and can provide higher levels of many ecosystem services [21,22]. The main reason for this is niche complementarity, which can increase the utilization ratio of resources and reduce competition for resources [23]. Moreover, the advantages of species mixing are generally considered to be more notable at sites with poor conditions (e.g., nutrient-poor and dry sites) than those with better conditions [24].

Because of their special geographic location, being an ecotone of the central subtropical zone and the southern subtropical zone according to the China Vegetation Regionalization, the forest ecosystems of Nanling are particularly sensitive to climate variation. Therefore, the Nanling Mountains are extremely vulnerable to the probability of natural disasters, such as insect outbreaks, typhoons, landslides, flooding, and ice damage. Moreover, increasing trends in temperature, especially in winter, may have no benefit to forest productivity. Nevertheless, long-term meteorological records and forest inventory are insufficient in this region. In this context, tree-ring analysis is an extremely valuable tool for managing natural and planted forests that can be used to reconstruct recent climatic trends as indirect evidence of past climatic changes [25–29]. Previous reports have shown that even short disturbances, such as a dry period, ice damage, and warming winters, in evergreen forests may result in false annual ring boundaries [30–32]. In addition, enhanced forest productivity in mixed species stands was related to factors that influence ecosystem functioning, especially those factors influencing the differences between plant species, such as site conditions [33], shade tolerance [34], crown phenology, canopy structure [34], root depth [35], and the rhizosphere microbiome [36]. The influence of these interactions gives rise to the greatest uncertainties in estimating future forest productivity and biodiversity [37]. However, higher plant diversity that may result in greater ecosystem productivity has aroused widespread attention and disagreement over recent decades. Some previous research

showed that mixed species composition did not increase aboveground productivity in the study area [37,38].

Inevitably, local climate change can be affected by large-scale climate driving factors, such as the Southern Oscillation Index (SOI) from the Stand Tahiti-Stand Darwin sea level pressure and the sea surface temperature (SST) of the Pacific and Indian Oceans [39,40]. Many previous studies also indicated that these large-scale climatic phenomena can directly or indirectly affect tree growth by modulating the local climate [41–45]. Hence, understanding the correlation between stand aboveground productivity and large-scale climate variation (e.g., multivariate ENSO index, MEI) is as important as the relationship between ANPP and local climatic conditions. The main aims of this study were (1) to explore whether the tree growth of different stand types in a common region can be associated with local climate variables, including monthly maximum temperature (Tmax), monthly minimum temperature (Tmin), precipitation (PRE), streamflow (FLO), and the number of dry days (DD), and (2) to assess the relationship between the aboveground productivity of stands and large-scale climatic drivers (MEI).

## 2. Materials and Methods

### 2.1. Research Site

The research site was located in the Jiulianshan National Nature Reserve, covering 134.1 km$^2$ of subtropical forest in the eastern Nanling Mountains of South China (Figure 1). Biodiversity in the reserve is abundant, and there are approximately 2796 species of higher plants belonging to 1112 genera and 297 families [46]. Of these species, the majority belong to the families of Fagaceae, Lauraceae, Magnoliaceae, Theaceae, Ericaceae, Leguminosae, and Ebenaceae. The lush woody vegetation was composed of a variety of tree and shrub species that resulted in a huge complexity for estimating the stand aboveground biomass and productivity. The soil types are typical upland yellowish red soil. To explore the climatic sensitivity of the subtropical mountain net primary productivity, four stand types differing in disturbance history were selected from the Xia Gongtang Ecosystem Positioning Research Station (Figure 1). The primary forest (Pri) covers an area of 4283.5 ha and was not influenced by any major disturbance for over 100 years except for some woods being used for mushroom cultivation [38]. In the primary subtropical forest, *Castanopsis carlesii* Hayata (Fagaceae), *C. eyrei* Tutch (Fagaceae), *C. fargesii* Franch (Fagaceae), *C. faberi* Hance (Fagaceae), *C. fordii* Hance (Fagaceae), *Schima superba* Gardn et Champ (Theaceae), *Machilus pingii* Cheng ex Yang (Lauraceae), *Manglietia fordiana* Oliv (Magnoliaceae), *Neolitsea aurata* Koidz (Lauraceae), and *N. chuii* Merr (Lauraceae) were the dominant and/or codominant species; *Diospyros morrisiana* Hance (Ebenaceae), *Litsea elongate* Benth et Hook (Lauraceae), *Michelia chapensis* Dandy (Magnoliaceae), *Rhododendron moulmainense* Hook (Ericaceae), and *Taxus chinensis* Rehd. var. *mairei* Cheng et L. K. Fu (Taxaceae) were also common. For the other three forest types, most trees >7 cm in diameter at breast height (DBH) were logged before 1988 (personal communication). By the late 1980s, all the harvested areas had been replanted using coniferous species (for more details see [46,47]). The secondary forest (Sec) was logged before the 1980s and thus is relatively poor in species richness, but all seedlings or saplings were reserved to cultivate seed production stands. The mixed forest (Mix) was incompletely harvested and planted with Chinese fir (*Cunninghamia lanceolata* (Lamb.) Hook.) in the spring of 1979 after logging. It has been developed into a representative conifer and broadleaf mixed forest because a few natural broadleaf tree species (mentioned above) were reserved (ca. 10% of the initial aboveground biomass [48]) after cutting. The single-species forest (Sin) was completely cleared before 1988 and planted with Chinese fir in the late 1980s. After the initial planting, a forest tending approach was implemented for 1 and 5 years. More supporting information on tree species composition, stand structure, hydrological characteristics, and soil parameters can be found in previous publications conducted at the same research site [37,48,49].

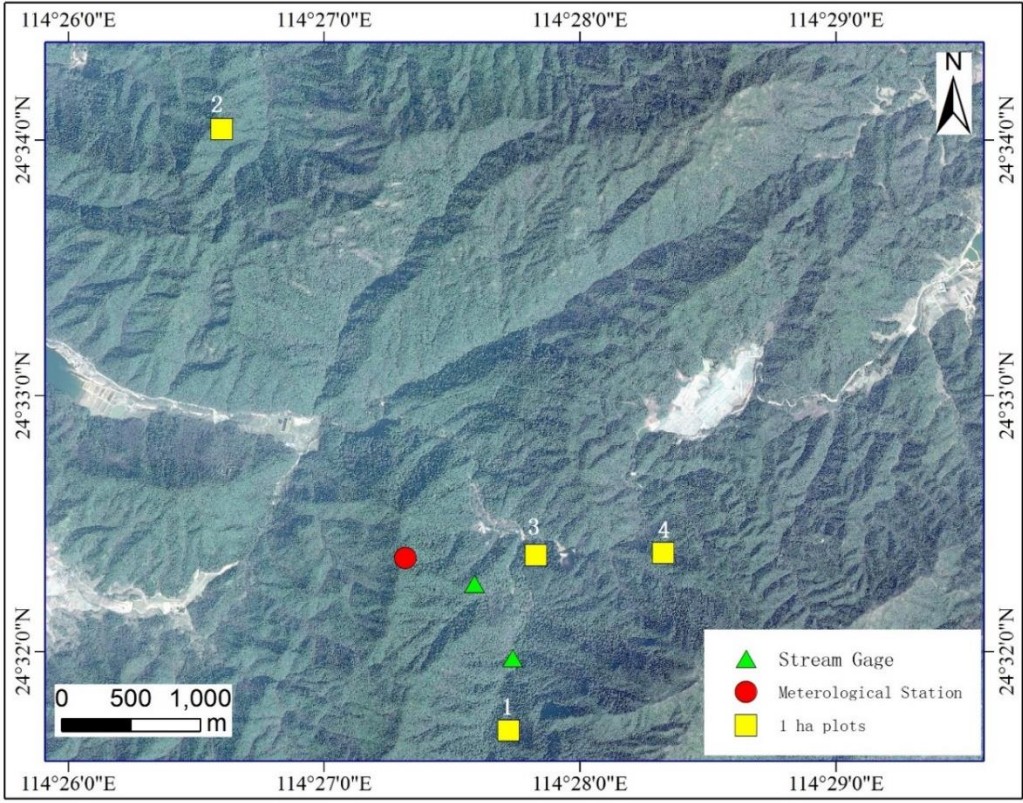

**Figure 1.** Map of the study site. 1, 2, 3, and 4 represent primary, secondary, mixed, and single-species stand plots, respectively.

Meteorological data (air temperature, maximum air temperature, minimum air temperature, and precipitation) measured by a ground meteorological station (Figure 1) were available to inform the calculation of the correlation with stand aboveground productivities. Because the meteorological data available for the study area covers only a short period (1990–2015), the climate data before 1990 were obtained from the National Meteorological Information Center (http://data.cma.cn/). For this reason, interpolated climatic data derived from two nearby climate datasets were used. Under the research period (1981–2015), the average annual temperature was $17.1 \pm 0.8$ °C, the average annual precipitation was $1816.3 \pm 344$ mm, and the mean number of dry days was $204 \pm 15$ d year$^{-1}$. The monthly minimum temperature averaged $9.9 \pm 2.5$ °C, ranging from $-0.49$ °C (January) to $20.4$ °C (July), and the monthly maximum temperature averaged $25.9.8 \pm 2.9$ °C, ranging from $19.0$ °C (January) to $31.4$ °C (July).

Likewise, streamflow data were available from a catchment in the study area, but these data also cover only a short period from 2000 to 2015. Before 2000, the monthly streamflow can be obtained from the following equation [49]:

$$y = 0.4321x + 16.2610 \ (R^2 = 0.85) \tag{1}$$

where y is the streamflow and x stands for the monthly precipitation. The mean monthly streamflow was $83.3 \pm 14.7$ mm, ranging from $35.8$ mm (November) to $160.0$ mm (June), and the annual surface runoff was approximately 14.3% of the total annual streamflow.

## 2.2. Research Design

In July 2015, three 1-ha fixed plots for primary forest, secondary forest, and mixed forest types were set up in the research site. The plot size for the three forest types was $100 \times 100$ m and was divided into subplots of $10 \times 10$ m each. Because of the extremely similar stand structure, 10 circular plots (200 m$^2$

per plot) were established in the single-species forest site (Figure 1). Four sites were approximately 800–3000 m from each other. All plots were similar in topography and site conditions [37].

In each plot, all plants ≥1 cm DBH were systematically recorded to species with the exception of a few that were identified to the genus level because it was difficult to identify them to species in the field without flowers or fruits. To estimate the aboveground productivity of the four stands from 1981 to 2015 (1992–2015 for single-species forests), 58, 61, and 37 one-centimeter diameter increment cores were collected from the secondary forest, mixed forest, and single-species stand plots, respectively. Given the complexity of the primary forest, a total of 232 one-centimeter diameter increment cores were collected from more 30 dominant tree species (for more details see [46]).

Ring widths were measured using winDendro software. Images were displayed in winDendro, and ring boundaries were manually determined. Because the stand ages of the secondary, mixed, and single-species stand plots were known in advance, it was easy to visually determine the ring width chronology. For the tree rings of the primary forest plot, the ring width quality control was crossdated using COFECHA [50]. Using the tree ring analysis, the DBH (diameter at breast high) in any year can be developed based on the ring width variation of each year [37,48]. The total aboveground biomass increment in Mg ha$^{-1}$ year$^{-1}$ is then calculated using the following allometric equation for tree layers [37]:

$$W_{AGB} = 0.1060D^{2.4520} (n = 33, R^2 = 0.99) \tag{2}$$

where $W_{AGB}$ represents the dry weight (kg) of the individuals whose DBH ≥5 cm and ≤50.2 cm. Similarly, the stem and canopy (including branch and leaf) aboveground biomass can be obtained from the following equations developed at the same study site (in press):

$$W_{LFB} = 0.0244D^{1.8827} \left( n = 33, R^2 = 0.95 \right) \tag{3}$$

$$W_{BRB} = 0.0132D^{2.5101} \left( n = 33, R^2 = 0.92 \right) \tag{4}$$

$$W_{BLB} = 0.0748D^{2.4780} \left( n = 33, R^2 = 0.99 \right) \tag{5}$$

where $W_{LFB}$, $W_{BRB}$, and $W_{BLB}$ represent the leaf, branch, and stem aboveground biomass (dry weight, kg), respectively.

For estimation of the aboveground biomass of understory woody species (<5 cm DBH), a total of 39 woody species with 746 individuals were destructively sampled to develop the allometric equation (ln $(w_{agb})$ = −0.312 + 0.880ln$d^2h$, where $w_{agb}$ is the dry weight (kg), $d$ is the ground diameter, and $h$ is tree high). In each plot, the above ground biomass of all seedlings or saplings (1 cm ≤ DBH < 5 cm) was calculated and divided by 5 as the annual net primary productivity of the shrub layer [51]. For the primary forest, secondary forest, mixed forest, and single-species forest plots, the values of aboveground net primary productivity were 0.9, 0.4, 0.9, and 0.05 Mg ha$^{-1}$ year$^{-1}$, respectively. The herb layer aboveground biomass of each plot was determined by a harvesting method, and the total herb aboveground biomass was divided by 3 to determine the herb layer aboveground net primary productivity [51]. The net primary productivity of the herb layer was 0.4, 0.1, 0.2, and 0.04 Mg ha$^{-1}$ year$^{-1}$ for primary forest, secondary forest, mixed forest, and single-species forest plots, respectively.

In this research, the stem aboveground biomass for each year was defined as the stem/wood productivity (WP), and the branch and leaf aboveground biomass was defined as the canopy productivity (CP). The total aboveground net primary productivity (ANPP) for each forest type includes the tree layer, shrub layer, and herb layer biomass. Due to the limited research funding and time, litter falls were not considered in this study.

To explore the sensitivity of the seasonal variation of ANPP, WP, and CP to climate change, the annual aboveground biomass increment of each forest type was distributed among the 12 months of a year according to the seasonal variation of net primary productivity in China [52,53]. The monthly net primary productivity of the subtropical evergreen broadleaf forest, mixed forest, and coniferous

forest was calculated using monthly 1 km AVHRR (Advanced Very High Resolution Radiometer) NDVI (Normalized Difference Vegetation Index) data, climate data, a vegetation type map, and a soil texture map. The results obtained were compared with ground-observation and Miami model results, showing that the results using remote sensing data are more accurate [52,53]. The monthly distribution of the annual net primary productivity for the subtropical evergreen broadleaf forest, mixed forest, and coniferous forest types can be found in Table 1.

**Table 1.** The percentage of annual net primary productivity, shown monthly, for evergreen broadleaf forests (EBLF), evergreen coniferous forests (ECF), and mixed broadleaf-conifer forests (MBCF) located in South China (recalculated from [53]). In this study, the EBLF includes primary and secondary forests, the ECF includes single-species forests, and the MBCF includes the mixed forest.

| Month | 1 | 2 | 3 | 4 | 5 | 6 | 7 | 8 | 9 | 10 | 11 | 12 |
|---|---|---|---|---|---|---|---|---|---|---|---|---|
| $P_{EBLF}$ (%) | 2.4 | 3.1 | 6.7 | 5.4 | 7.9 | 12.9 | 14.1 | 15.9 | 14.1 | 7.8 | 5.8 | 3.7 |
| $P_{ECF}$ (%) | 1.1 | 1.5 | 3.8 | 4.9 | 8.7 | 13.8 | 19.2 | 17.0 | 16.0 | 7.7 | 4.1 | 2.1 |
| $P_{MBCF}$ (%) | 1.8 | 2.3 | 5.3 | 5.2 | 8.3 | 13.3 | 16.7 | 16.5 | 15.1 | 7.7 | 5.0 | 2.9 |

### 2.3. Multivariate ENSO Index

The multivariate ENSO index (MEI) was downloaded from the National Oceanic and Atmospheric Administration (NOAA) (https://www.esrl.noaa.gov/psd/enso/mei/table.html). As a multivariate monitor of the ENSO signal based on sea level pressure, surface temperature, surface air temperature, and cloudiness over the tropical Pacific Ocean, the MEI is separately computed for each of the 12 sliding bimonthly seasons and is the first principal component of all six observed fields combined [54,55]. The MEI integrates more information than other indices and reflects the nature of the coupled ocean-atmosphere system better than either component, and it is less vulnerable to occasional data glitches in the monthly update cycles. Negative MEI values stand for the cold ENSO phase related to heavy precipitation events (also known as La Niña), while positive values of the MEI stand for the warm ENSO phase associated with droughts in the research region (El Niño).

Thus far (1950–2017), MEI rankings from 1-14 indicate La Niña, while those from 55–68 indicate El Nino. Finally, the strong ENSO events may be defined by the top 7 rankings, such as 1–7 for strong La Niña and 62–68 for strong EI Niño events. More details can be found on the MEI website (http://www.esrl.noaa.gov/psd/enso/mei/).

### 2.4. Climate-Productivity Analysis

Climate effects on forest productivity were evaluated by relating the stand aboveground productivity with the daily records of meteorological parameters, i.e., precipitation (PRE), streamflow (FLO), mean maximum and minimum annual temperatures (MTmax and MTmin), and the number of dry days (DD). From January 1981 to December 2015, the monthly total aboveground net primary productivity (ANPP), wood productivity (WP), canopy productivity (CP), precipitation (PRE), streamflow (FLO), and the number of dry days (DD) were computed as well as the average monthly minimum and maximum temperatures (Tmin and Tmax). All raw data were subjected to a normal distribution test, and non-normal distribution data were transformed to follow a normal distribution using logarithmic transformation. For each forest type, all the correlation analyses between the stand aboveground productivity and climatic variables were calculated using SPSS version 24.0 [56].

To explore the effects of climate change on subtropical mountain aboveground net primary productivity, the annual sums and the monthly averages of climatic and productivity parameters were calculated by separating the patterns of interannual and seasonal variations [45]. Likewise, the relationship between large-scale integrative climatic parameters (e.g., MEI) and local climate variation (temperature, precipitation, streamflow, and the number of dry days) and their direct impacts on forest productivity were analyzed. Spearman's correlation coefficients were applied to test the

relationships between the monthly stand aboveground productivity and monthly climatic variables for each forest type.

The annual sums or averages of climatic variables during the study period of 1981–2015 were computed, and the significance relationship of MEI with the annual average precipitation, streamflow, number of dry days (DD), maximum temperature, and minimum temperature were tested using correlation analysis. Using multiple linear regression models, the monthly average values of climatic and productivity variables were calculated from the period of 1981–2015. A canonical correlation analysis (CCA) was used to assess and enhance the visualizations between climatic and productivity parameters.

Based on the seasonal decomposition method, the climatic and forest productivity parameters were decomposed into a trend component, seasonal component, and residual component to verify the regularities in interannual and seasonal fluctuations [45]. Linear mixed effects models (with interannual trend as a fixed factor and seasonal variation as a random effect) were used to investigate the significant trends among interannual forest productivity and climatic variables during the study period of 1981–2015.

The climatic and productivity data were detrended by subtracting the monthly means from each month's actual values to derive the interannual anomalies and subtracting the annual means from each month's actual values to derive the seasonal anomalies [45,57]. One-way analysis of variance was used to determine whether a significant difference emerged in the interannual detrended data and the seasonal detrended data. Lagged correlation analysis was applied to obtain the highest correlation coefficients between seasonally detrended climatic and productivity variables at different time lags to determine significant relationships among the investigated parameters. Such a climatic lag on tree growth has been validated in temperate and tropical wood stands [45,58,59]. To reflect the significant climatic factors driving subtropical mountain ANPP, the Spearman correlation coefficient between the forest productivity and climatic variables was plotted.

## 3. Results

### 3.1. Influence of MEI on Local Climate

During the research period (1981–2015), the MEI rankings showed that there were four significant ENSO transition periods: 1986–1989, 1995–1999, 2006–2008, and 2009–2011 (Figure 2). In each of the four ENSO transition periods, the relationships between annual precipitation, streamflow, mean maximum annual temperature, and the number of dry days and the MEI were remarkably similar in trend (Figure 3). The mean maximum annual temperature increased by 2.5, 0.9, 6.5, and 0.9 °C during the 1986/1988, 1997/1998, 2006/2007, and 2009/2010 EI Niño periods, respectively. Over the same transition periods, the annual precipitation decreased from 1765.6 mm to 1420.6 mm, from 1844 mm to 1635 mm, from 2358.6 mm to 1809.3 mm, and from 1821.9 mm to 1416.1 mm during 1986/1988, 1997/1998, 2006/2007, and 2009/2010 EI Niño events and subsequently reached peak values of 1875.3 mm, 2214.2 mm, 1821.9 mm, and 2355 mm during the 1988/1989, 1998/2001, 2007/2008, and 2010/2011 La Niña events, respectively (Figures 3 and 4). Likewise, the annual streamflow and number of dry days were similar to the trend of the annual precipitation during the four ENSO transition periods.

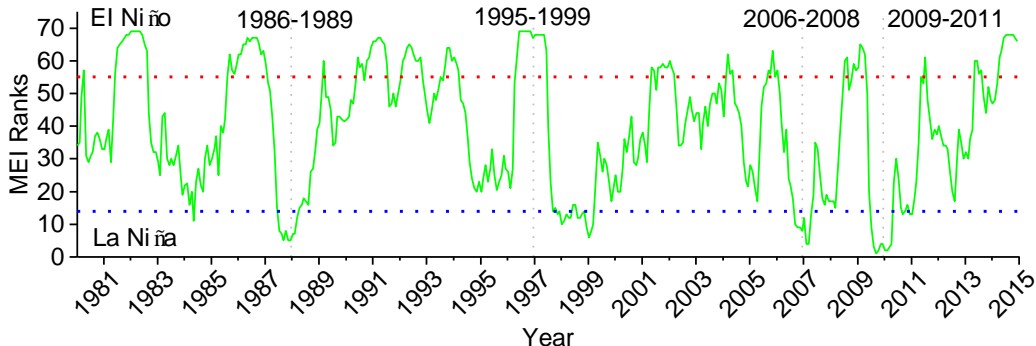

**Figure 2.** MEI (Multivariate ENSO Index) ranks during the period of 1981–2015. Generally, MEI ranks from 1–14 (the blue line represents a ranking of 14) represent La Niña, whereas 55–68 (the rank value at red line is 55) represent EI Niño.

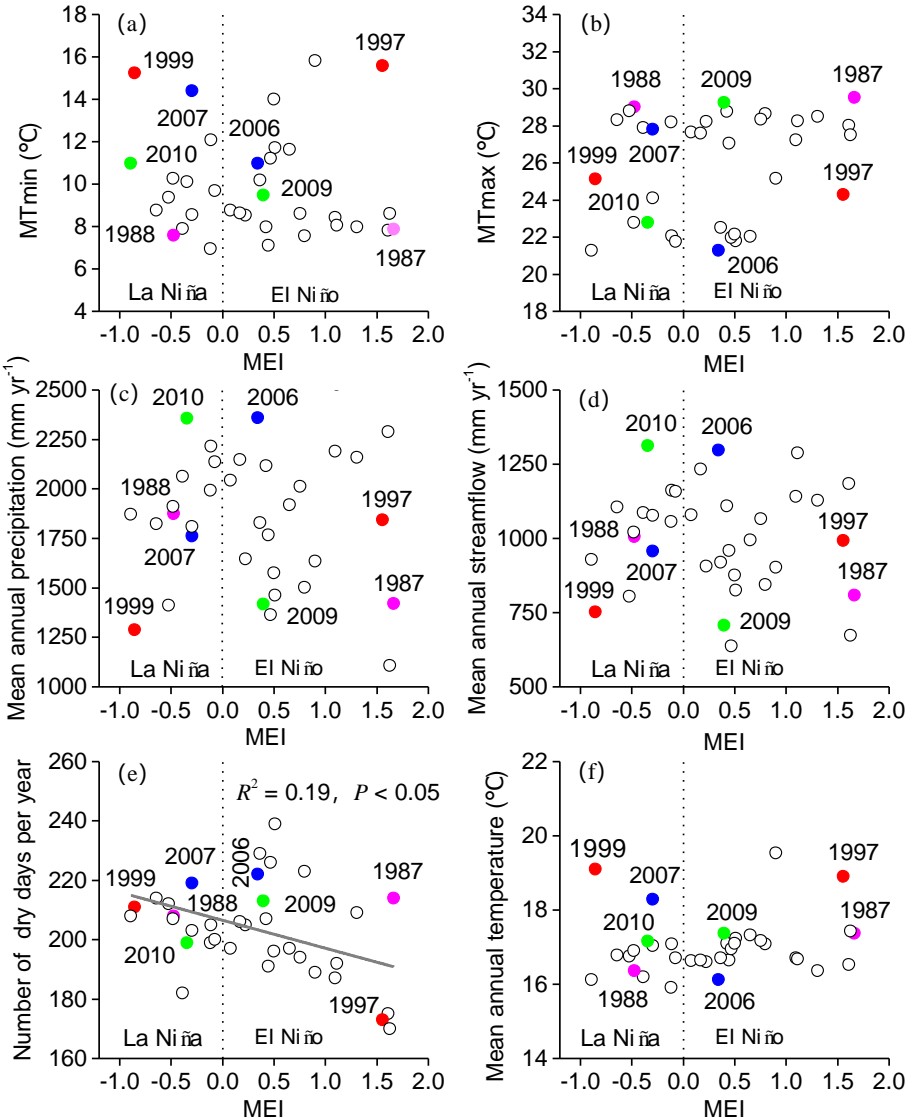

**Figure 3.** Relationship between MEI and local climatic variables: mean minimum and maximum temperature (MTmin (**a**) and MTmax (**b**)), mean annual precipitation (**c**) and streamflow (**d**), and the number of dry days (**e**), mean annual temperature (**f**). MEI represents the multivariate ENSO index. The $R^2$ and *p* value represents the significant level.

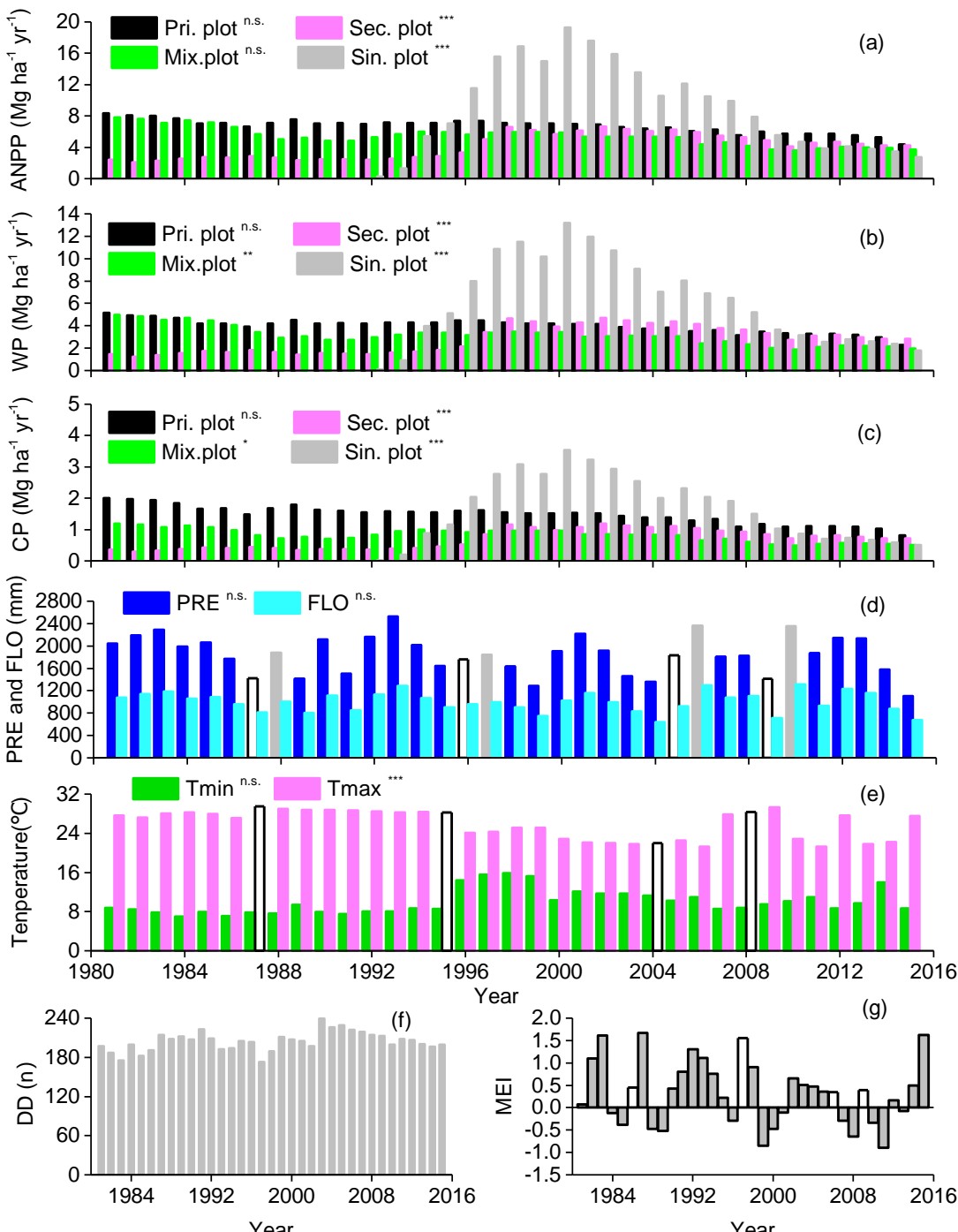

**Figure 4.** Interannual fluctuation in the annual sums of productivity (aboveground net primary productivity—ANPP (**a**), wood productivity—WP (**b**), and canopy productivity—CP (**c**)) and climatic (precipitation—PRE and streamflow—FLO (**d**), Tmin and Tmax (**e**), the number of dry days—DD (**f**) and MEI (**g**)) variables during the study period of 1981–2013 for primary (Pri, black bars), secondary (Sec, purple bars), mixed (Mix, green bars), and single-species (Sin, gray bars) forests located in South China. The significant differences in interannual changes were computed from seasonal detrended variables using one-way ANOVA (Table S2 in the Supplementary Materials) and are marked by asterisks (the asterisks "***", "**", and "*" represent the significance levels of 0.001, 0.01, and 0.05, respectively, and "n.s." represents non-significant levels). In addition, bars in the low four diagrams in climate variables (MEI, Tmax, Tmin, PRE, and FLO) stand for respective climate anomalies during the El Niño period (white bars) as well as next La Niña period (grey bars).

*3.2. Correlations between Forest Aboveground Productivity and Climatic Drivers*

　　　Spearman's correlation coefficients and significant values indicated that there existed a significant relationship between the various climatic parameters and productivity variables (Table S1 in Supplementary Materials). All the productivity variables were significantly and strongly correlated with the minimum and maximum temperatures, underlying the importance of extreme temperature for tree growth in subtropical Chinese mountains. Specifically, streamflow (FLO) was more highly correlated than precipitation (PRE) with all productivity variables (Table S1 in the Supplementary Materials).

　　　The responses of ANPP, WP, and CP to climate changes showed similar responses to seasonal and interannual changes (Figures 4 and 5, Tables S2 and S3 in the Supplementary Materials). However, the WP for the single-species forest plot displayed a significant interannual response in 1998, 2000, 2005, and 2009 (Figure 4), but did not have a strong pattern of seasonal climate change (Figure 5 and Table S2 in the Supplementary Materials). The seasonal climate variation was also influenced by ENSO events (Figure 5). For example, the monthly rainfall and streamflow had a significant reduction, and the monthly average maximum temperature had a significant increase during the time of El Niño (Figure 5).

　　　The visualizations showed that there was a close relationship between interannual productivity and climatic variables (Figure 6 and Table S4 in the Supplementary Materials). The variables describing the primary and mixed forest annual productivity and its components (ANPP Pri, WP Pri, CP Pri, ANPP Mix, WP Mix, and CP Mix) were positively correlated with the mean maximum annual temperature (MTmax), annual precipitation (PRE), annual streamflow (PRE), and MEI, but negatively correlated with the number of dry days (DD) and mean minimum annual temperature (MTmin). The secondary and single-species forest productivity and component variables were positively correlated with precipitation-free days (DD) and MTmin, and inversely correlated with MTmax, PRE, FLO, and MEI during the study period of 1981–2015.

　　　Multivariate linear regression analysis showed that stand aboveground productivity had a strong response to climatic variations (Table 2). For all stand types in the study region, there were no significant relationships between productivity and the climatic parameters of DD and MEI; Tmin was positively correlated to ANPP and each of its components for the four forest types. Tmax and PRE were positively correlated with the primary and mixed forest ANPP and its components, but negatively correlated with the secondary and single-species forest ANPP and its components. However, streamflow (FLO) was negatively associated with the primary and mixed forest ANPP and its components and opposed to secondary forest WP, CP, and single-species forest ANPP and its components (Table 2).

　　　In addition, the time series decomposition was carried out by anatomizing seasonal and interannual changes and investigating the detrended seasonal and interannual signals of stand aboveground productivity and climatic parameters (Figure 7). The time series analysis showed that interannual differences in the fluctuation range of climatic variables were significantly correlated with the El Niño/Southern Oscillation (ENSO) phenomenon (Table S4 in the Supplementary Materials).

　　　The Multivariate ENSO Index (MEI) dropped over the period of 1984–1986, 1994–1996, 1998–2000, and 2006–2008 and reached peak values in late 1987, 1997, 1999, and 2009, while the detrended interannual fluctuation of precipitation (PRE) and streamflow (FLO) displayed opposed extreme values in 1987, 1997, 1999, and 2009, respectively. However, the detrended interannual variation of dry days (DD) was prolonged to the corresponding next year and thus meant a lagged time of drought over the ENSO transition period. Likewise, the mean maximum annual temperature increased dramatically by 2.5, 0.9, 6.5, and 0.9 °C, and the interannual trend of precipitation (PRE)/streamflow (FLO) decreased by 345/149.1 mm, 208.6/90.1 mm, 549.3/221.3 mm, and 405.8/397.4 mm during the four ENSO transition periods, respectively. It is noteworthy that MEI seasonally detrended interannual signals were only not significant, and PRE were not extremely significant (Table S4 in the Supplementary Materials). At the same time, the seasonally detrended interannual variation of stand aboveground productivity and its components expressed fluctuations in response to climate changes (Figure 7 and Table S4 in the Supplementary Materials). For example, there was an obvious decline

in ANPP, WP, and CP of secondary and single-species forest plots during the 1998–2001 La Niña period and a slight increase in mixed, single-species, and secondary forest aboveground productivity during the 2002–2003 and 2006–2007 EI Niño periods, respectively (Figure 7). After the 1986–1988 EI Niño events, the primary forest aboveground productivity slightly increased as a prolonged response to climate change (Figure 7).

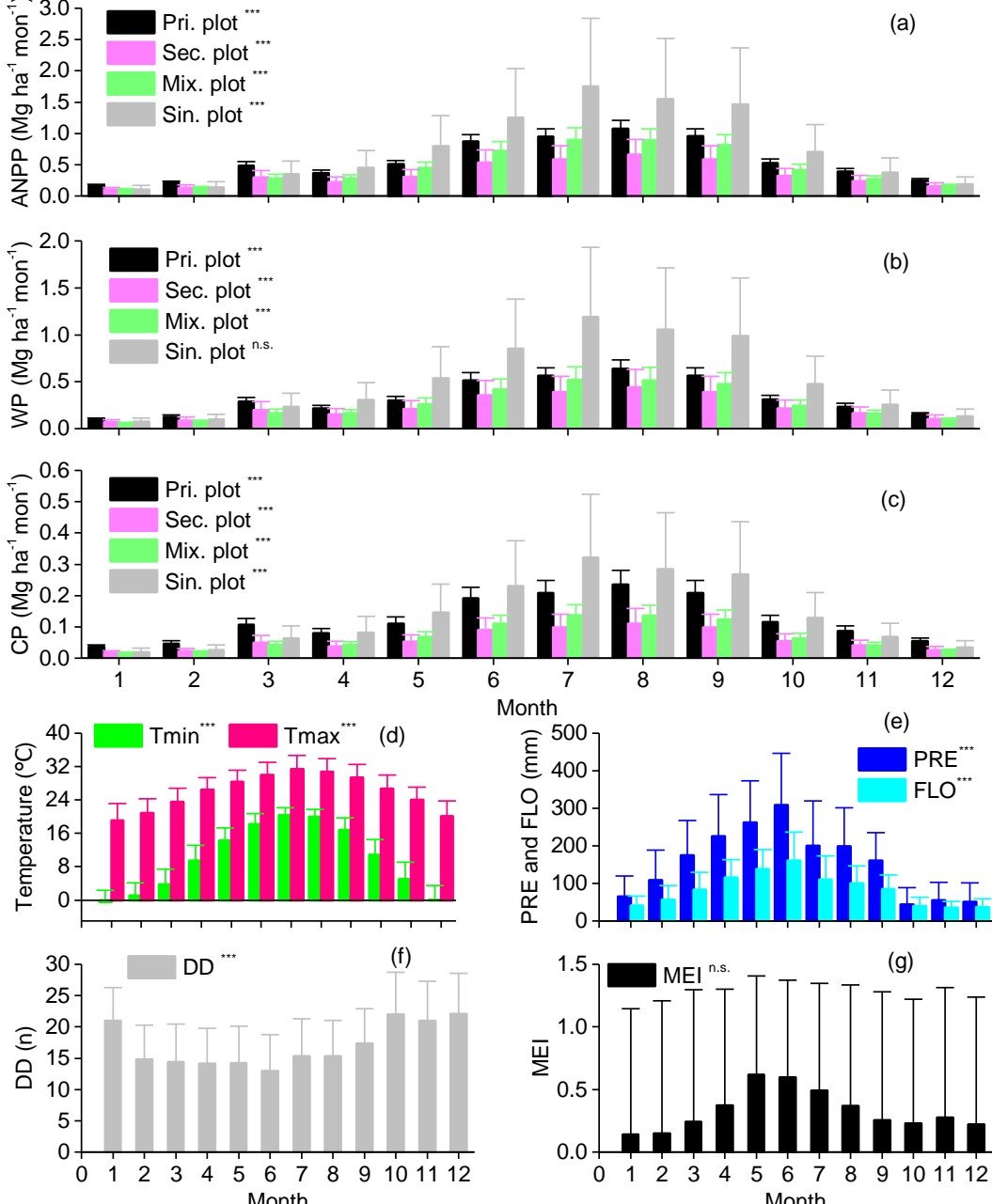

**Figure 5.** The monthly average productivity (ANPP (**a**), WP (**b**), and CP (**c**)) and climatic (Tmin and Tmax (**d**), PRE and FLO (**e**), DD (**f**) and MEI (**g**)) variable variation with the seasons during the study period of 1981-2013 for primary (Pri, black bars), secondary (Sec, purple bars), mixed (Mix, green bars), and single-species (Sin, gray bars) forests located in South China. The bars are the monthly mean values, and the solid lines represent the standard errors. The significant differences in seasonal variation were estimated from interannual detrended variables using one-way ANOVA (Table S3 in the supplementary martials) and are marked by asterisks (the asterisks "***", "**", and "*" represent the significance levels of 0.001, 0.01, and 0.05, respectively, and "n.s." represents non-significant levels).

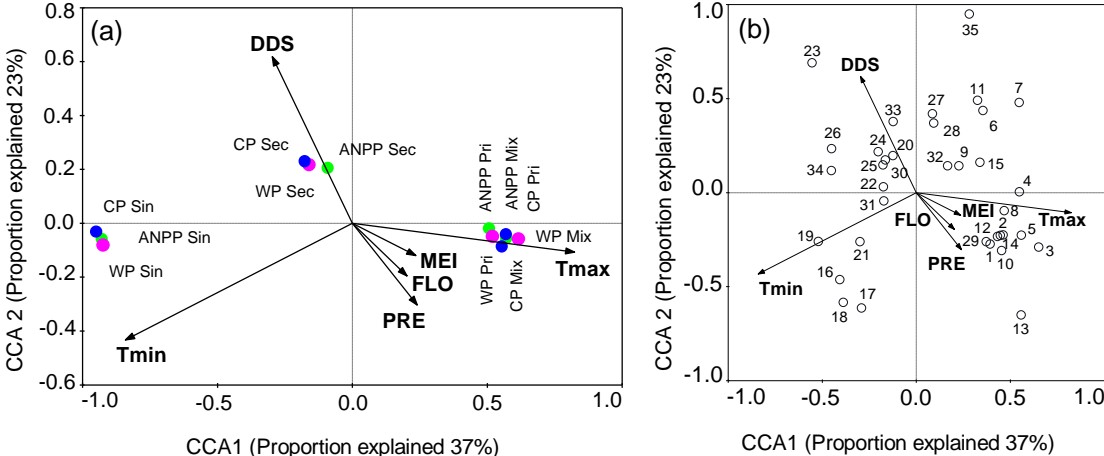

**Figure 6.** The results of canonical correspondence analysis (CCA) showing (**a**) the relation between the average annual climatic variables and productivity variables and (**b**) the relationship of the annual average climatic variables to the annual productivity variables during the study period of 1981–2015. The monthly average climatic variables included the multivariate ENSO index (MEI), minimum temperature (Tmin), maximum temperature (Tmax), precipitation (PRE), streamflow (FLO), and the number of dry days (DD). The mean annual productivity variables included aboveground net primary productivity (ANPP), wood/stem productivity (WP), and canopy productivity (CP) for primary (Pri), secondary (Sec), mixed (Mix), and single-species (Sin) forests located in South China. The numbers 1–35 in (b) represents the period of 1981–2015.

Finally, there existed notable and lagged effects between seasonally detrended subtropical mountain aboveground productivity and climatic variables by using lagged cross correlation analysis (Table S5 in the Supplementary Materials). The analysis indicated that not all the seasonally detrended stand aboveground productivities were significantly lagged with the seasonally detrended climatic variables precipitation (PRE), streamflow (FLO), and rain-free days (DD) (Figure 8). In contrast, the interannual variation of the multivariate ENSO index (MEI), and the mean maximum (Tmax) and minimum (Tmin) temperatures exhibited significant and strong lag effects (between 0 and 36 months) for the overwhelming majority of subtropical mountain aboveground productivity and its components (Figure 8 and Table S5 in the Supplementary Materials). After sustaining a slight and brief increase, MEI clearly decreased stand aboveground productivity and its components among the first 18 months under the warm and drought conditions of the El Niño phenomenon, and ANPP, WP, and CP reached a peak value in the next 30 months as a prolonged response (between 19 and 30 months) to the cool and wet La Niña phenomenon. Similarly, Tmax significantly reduced subtropical mountain aboveground productivity (except single-species forest productivity) over the first 12 months, and the productivity values peaked 25 months after the abnormal drought weather because of lagged effects of the La Niña phenomenon. It is noteworthy that Tmin exhibited a weak and positive effect on ANPP and its components, and DD expressed a similar lagged effect on the productivity compared with the effects of Tmax on the drought and moisture-related anomalies.

Interestingly, FLO obviously increased the ANPP and CP of the primary forest plot, but did not significantly influence other stand types. Thus, the present result also revealed the sensitive response of subtropical mountain forests to large-scale climatic anomalies, such as the El Niño/Southern Oscillation (ENSO) phenomenon.

**Table 2.** Results of multiple linear regression models exploring the influences of the monthly mean climatic variables on subtropical ANPP and its components (WP and CP). MEI is the multivariate ENSO index, Tmin is the minimum temperature, Tmax is the maximum temperature, PRE is the precipitation, FLO is the streamflow, and DD represents the number of dry days.

| | ANPP | | | | | | | |
|---|---|---|---|---|---|---|---|---|
| | **Primary Forest** | | **Secondary Forest** | | **Mixed Forest** | | **Single-Species Forest** | |
| **Variation** | **Estimates** | **p Value** | **Estimates** | **p Value** | **Estimates** | **p Value** | **Estimates** | **p Value** |
| Constant | −0.25520 | 0.000 | 0.22554 | 0.000 | −0.33853 | 0.000 | 0.19981 | 0.035 |
| MEI | −0.00850 | 0.276 | −0.00876 | 0.249 | 0.00079 | 0.921 | −0.04457 | 0.213 |
| Tmin | 0.02359 | 0.000 | 0.02653 | 0.000 | 0.02242 | 0.000 | 0.07922 | 0.000 |
| Tmax | 0.02206 | 0.000 | −0.00695 | 0.000 | 0.02210 | 0.000 | −0.01006 | 0.228 |
| PRE | 0.00063 | 0.003 | −0.00002 | 0.909 | 0.00045 | 0.038 | −0.00063 | 0.436 |
| FLO | −0.00127 | 0.003 | −0.00001 | 0.982 | −0.00100 | 0.022 | 0.00079 | 0.622 |
| DD | 0.00126 | 0.319 | 0.00208 | 0.091 | 0.00024 | 0.852 | −0.00086 | 0.872 |
| $R^2$ | 0.879 | | 0.798 | | 0.866 | | 0.725 | |
| p value | 0.000 | | 0.000 | | 0.000 | | 0.000 | |
| | **Wood Production** | | | | | | | |
| | **Primary Forest** | | **Secondary Forest** | | **Mixed Forest** | | **Single-Species Forest** | |
| **Variation** | **Estimates** | **p Value** | **Estimates** | **p Value** | **Estimates** | **p Value** | **Estimates** | **p Value** |
| Constant | −0.15887 | 0.000 | 0.20828 | 0.000 | −0.24726 | 0.000 | 0.17117 | 0.031 |
| MEI | −0.00498 | 0.306 | −0.00664 | 0.226 | 0.00122 | 0.819 | −0.02950 | 0.635 |
| Tmin | 0.01382 | 0.000 | 0.01847 | 0.000 | 0.01301 | 0.000 | 0.05370 | 0.000 |
| Tmax | 0.01346 | 0.000 | −0.00674 | 0.000 | 0.01434 | 0.000 | −0.00742 | 0.000 |
| PRE | 0.00040 | 0.002 | −0.00003 | 0.843 | 0.00029 | 0.043 | −0.00041 | 0.075 |
| FLO | −0.00080 | 0.002 | 0.00000 | 0.998 | −0.00063 | 0.028 | 0.00045 | 0.104 |
| DD | 0.00071 | 0.364 | 0.00113 | 0.204 | 0.00057 | 0.512 | −0.00132 | 0.895 |
| $R^2$ | 0.870 | | 0.776 | | 0.845 | | 0.722 | |
| p value | 0.000 | | 0.000 | | 0.000 | | 0.000 | |
| | **Canopy Production** | | | | | | | |
| | **Primary Forest** | | **Secondary Forest** | | **Mixed Forest** | | **Single-Species Forest** | |
| **Variation** | **Estimates** | **p Value** | **Estimates** | **p Value** | **Estimates** | **p Value** | **Estimates** | **p Value** |
| Constant | −0.07115 | 0.000 | 0.09443 | 0.000 | −0.04961 | 0.000 | 0.04743 | 0.030 |
| MEI | −0.00163 | 0.408 | −0.00224 | 0.189 | 0.00048 | 0.711 | −0.00900 | 0.175 |
| Tmin | 0.00476 | 0.000 | 0.00393 | 0.000 | 0.00350 | 0.000 | 0.01466 | 0.000 |
| Tmax | 0.00561 | 0.000 | −0.00233 | 0.000 | 0.00328 | 0.000 | −0.00243 | 0.117 |
| PRE | 0.00015 | 0.004 | −0.00002 | 0.631 | 0.00007 | 0.040 | −0.00013 | 0.403 |
| FLO | −0.00031 | 0.004 | 0.00003 | 0.774 | −0.00016 | 0.021 | 0.00017 | 0.563 |
| DD | 0.00024 | 0.455 | −0.00044 | 0.110 | 0.00002 | 0.935 | −0.00001 | 0.994 |
| $R^2$ | 0.851 | | 0.633 | | 0.854 | | 0.719 | |
| p value | 0.000 | | 0.000 | | 0.000 | | 0.000 | |

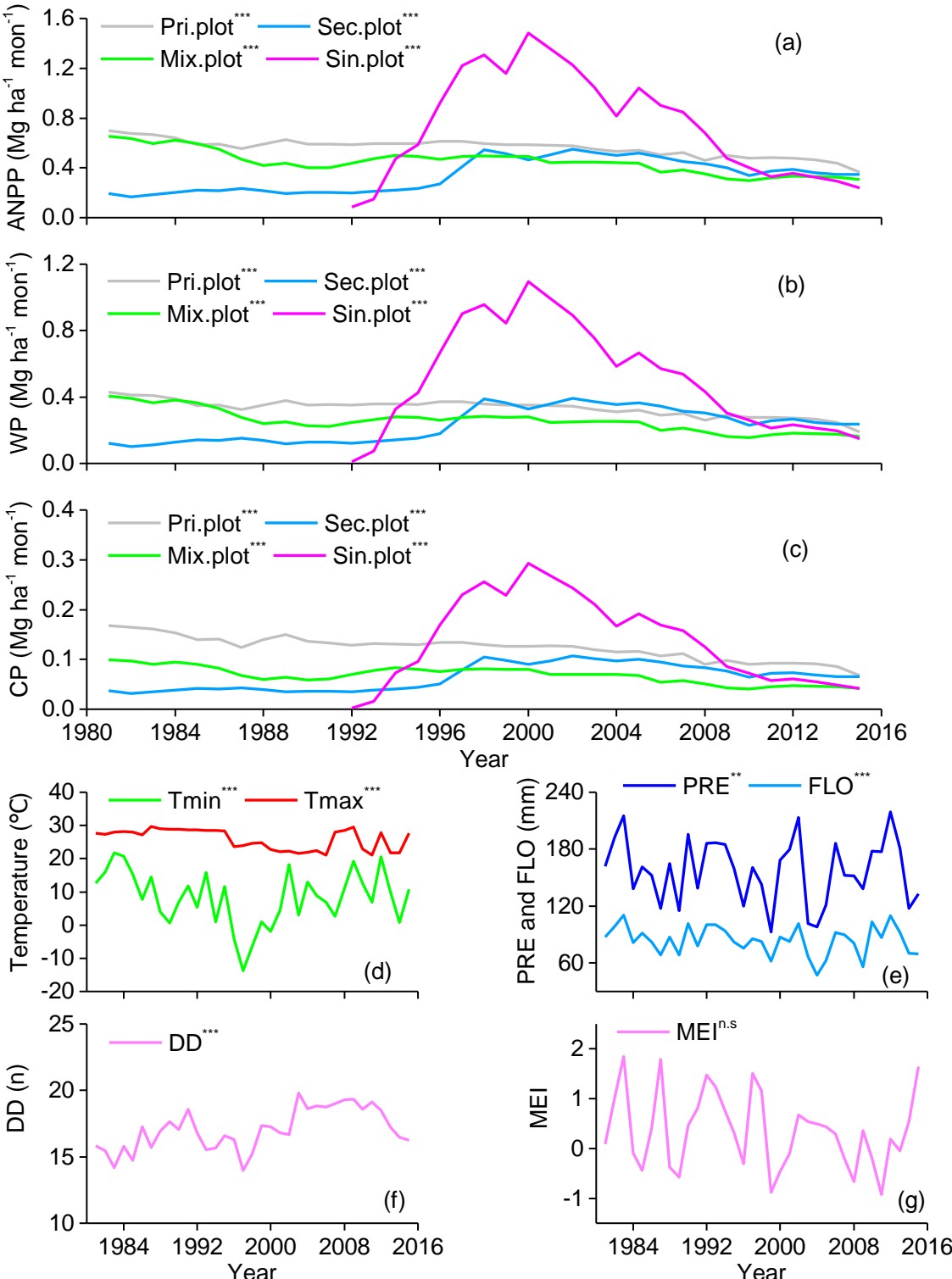

**Figure 7.** Time series of seasonally detrended climatic and productivity variables during the study period of 1981–2015. The seasonally detrended productivity variables included ANPP (**a**), WP (**b**), and CP (**c**) for primary (Pri), secondary (Sec), mixed (Mix), and single-species (Sin) forest plots in South China. The seasonally detrended climatic variables included the Tmin and Tmax (**d**), PRE and FLO (**e**), DD (**f**) and MEI (**g**). The significant differences in time series were computed using mixed linear models and are marked by asterisks ("***" represents the significance level of 0.001, "**" represents the significance level of 0.01 and "n.s" represents non-significant levels).

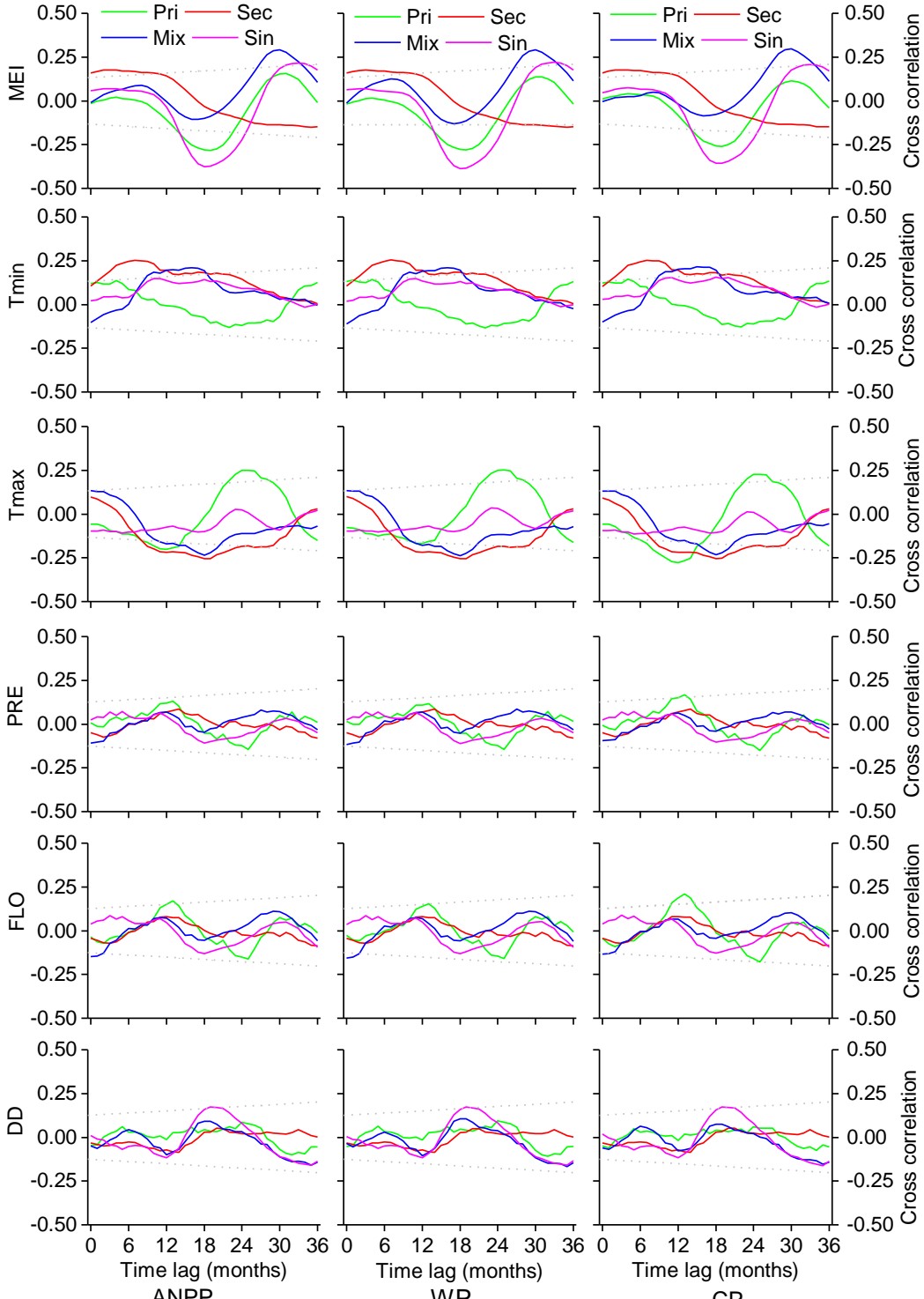

**Figure 8.** The lagged cross correlation values between climatic variables and productivity variables. The seasonally detrended climate variables included MEI (multivariate ENSO index), Tmin (minimum temperature), Tmax (maximum temperature), PRE (precipitation), FLO (streamflow), and DD (number of dry days). The seasonally detrended productivity variables included aboveground net primary productivity (ANPP), wood/stem productivity (WP), and canopy productivity (CP) for primary (Pri), secondary (Sec), mixed (Mix), and single-species (Sin) forest plots located in South China. The light dotted lines stand for the 95% confidence limits of significant lagged cross correlation coefficients between seasonal detrended climate and productivity variables ($p < 0.05$; more details can be seen in Table S5 in the supplementary martials).

## 4. Discussion

### 4.1. Correlation between Subtropical Mountain Productivity and Local Climatic Variation

Based on 35 years of observed climate and productivity data, this study sought to test whether local and regional climate changes can continue to affect aboveground productivity across different forest types in subtropical China. The correlation analysis showed that stand aboveground productivity is sensitive to local and large-scale climatic factors. On a local scale, further analysis indicated that the monthly minimum temperatures were positively correlated with the stand aboveground productivity in subtropical mountains, and monthly extreme high temperatures (Tmax) were negatively correlated with the stand aboveground productivity in subtropical regions (Table 2). The negative correlation between the monthly extreme high temperature and stand above ground productivity is consistent with a similar study in a tropical forest [60]. Likewise, the results of this study are similar to previous findings from many tropical study regions [59,61–63]. The results are also consistent with the conventional findings that the growth of trees located at lower latitudes is less influenced by the mean monthly temperature [64–66].

Possible explanations for this finding might include the following: (1) Lower temperatures in winter may restrict tree growth and hence can provide sufficient nutrients for the upcoming growing season; (2) higher temperatures in summer can increase evapotranspiration and thus leaf stomata are apt to close to reduce water loss, leading to lower $CO_2$ uptake and reduced carbon assimilation rates [67]; and (3) higher temperatures may trigger tree growth in winter and may lead to greater costs of plant maintenance in summer, thus resulting in nutrient deficiencies in the upcoming growing season [68,69]. A previous report showed an approximately two-fold increase in respiration rates when the air temperature increased by approximately 10 °C [70]. It was also reported that higher ambient temperatures can significantly increase the essential costs of maintaining plant tissues [71], although plants have the ability to adjust to such changes relatively quickly [72]. In summary, the maximum temperature can increase plant respiration in summer and nutrient consumption (by promoting tree growth) in winter and hence may lead to biomass accumulation reduction in subtropical forests. Minimum temperature can reduce the respiration of trees in summer and nutrient consumption (by inhibiting tree growth) in winter and thus may result in biomass accumulation increases during the following growing seasons.

Precipitation can significantly affect tree growth in many dry tropical forests [43,44,73–76], but the stand aboveground productivity is less affected by monthly and annual precipitation in the subtropical mountains. A previous study reported that the soil water content always exceeded 35% in the study site [77]. Hence, just as the mean monthly temperature, precipitation is not an essential limiting factor for tree growth at the study site.

There was a significantly negative relationship between stand productivity and the monthly streamflow of February and March (data not shown). It is likely that the larger discharge of flowing water could cause litter and nutrient loss and thus lead to a lack of nutrients and weakened photosynthesis in the upcoming growing season. However, there is a significant positive correlation between the stand productivity of primary and secondary stand plots and monthly stream flow data for June and July of the current year (data not shown). Although the higher channel runoff may lead to nutrient loss, the decomposition rate of mulching plants and the release rate of plant nutrients were faster in summer than in winter in the primary and secondary stands. In contrast, the larger stream flow represented increasing precipitation and means that the effects of cooling and dampening are favorable for tree growth in summer. In addition, the larger stream flows showed promising permeability of the soil, and the root growth of trees will not be limited. It was reported that soil respiration is highly sensitive to moisture restrictions over the short term, but not to associated temperature increases [70].

However, it is difficult to conclusively determine whether direct influences of climate on subtropical mountain productivity or rather inherent growth rhythms of plants regulate plant-growth allocation and thus control the response of tree growth to seasonal climatic variations. For example,

previous studies demonstrated that monthly leaf productivity was not synchronized with monthly stem/wood productivity because leaf productivity reached the peak values at the first of the dry season and stem/wood productivity did so at the early stage of the wet season [45,78]. Some researchers found that phenological rhythms are more likely to be involved in driving productivity allocation than direct effects of climatic factors [79–81]. These results were slightly different in comparison with this paper's findings. The results in the study showed that stand aboveground productivity and its components of primary and secondary forest plots increased gradually to a small peak at the late stage of the dry season (March) and peaked at the middle of the ensuing rains (between June and July), whereas stand aboveground productivity and its components of mixed and single-species forest plots peaked at the mid-to-late rainy season (from July to August) (Figure 5). One of the main reasons may be the differences in plant diversity among these vegetation communities. Some previous studies suggested that plant community composition and diversity can affect biomass allocation and increase fine root production; thus, trees in forests with higher species diversity can quickly take advantage of limited resources under adverse climatic situations [82–84]. In addition, the seasonal variation in light conditions can significantly affect the seasonal change in productivity, such that more biomass can be stored in trees in the cooler period with low light conditions and more energy may be consumed to increase the respiration of plants during the hotter season with higher light intensity [85,86].

### 4.2. Response of Subtropical Mountain Productivity to Large-Scale Climate Anomalies

Based on the consecutive data on subtropical stand aboveground productivity for primary, secondary, and mixed forests during the study period (1981–2015), the four relatively significant ENSO transition periods of 1986–1989, 1995–1999, 2006–2008 and 2009–2011 reduced the total stand aboveground productivity (ANPP) by an average of 5.7%, 3.0%, 2.4%, and 7.8% as a response to the increased extreme high temperature (Tmax) and drought during the EI Niño periods. Subsequently, the total productivity values increased by an average of 1.1%, 3.0%, 0.3%, and 8.6% because of the lagged effects after the cool and wet La Niña period. Over the 1995–1999, 2006–2008, and 2009–2011 ENSO transition periods, the single-species forest aboveground productivity declined by 11.1%, 5.9%, and 21% in response to decreased precipitation and elevated temperatures during the EI Niño episodes, and then the productivity values recovered by 28.9%, 0%, and 7.5% as a prolonged growth response after the La Niña episodes. Interestingly, the single-species forest productivity failed to recover growth after the 2008 ice damage, but recovered by only 7.5% during the next 2009–2011 ENSO transition period. It is also worth noting that the canopy productivity for all forest types recovered faster than wood/stem productivity after the wet La Niña episode during the study period (averages of 3.5% to 0.7%, 3.7% to 3.2%, 0.9% to 0.8%, and 8.0% to 7.8% during the four ENSO transition periods of 1986–1989, 1995–1999, 2006–2008, and 2009–2011, respectively). A recent study demonstrated that the canopy density was one of the most important driving variables of forest carbon stocks in subtropical forests [87].

In other words, large-scale climate anomalies can actually influence tree growth, but each forest type exhibited an individual pattern of response to climate events in the subtropical mountains. Actually, this research revealed that primary and mixed forest aboveground productivity and its components increased with maximum temperatures, precipitation, and streamflow, the secondary forest aboveground productivity increased with the number of precipitation-free days, and the single-species forest aboveground productivity was positively associated with minimum temperatures (Figure 5). Some recently published studies have underlined the important influences of climatic variables in regulating forest biomass and carbon storage [88–91]. In general, the productivity of subtropical forests was positively correlated with rainfall and temperature across different forest types because greater amounts of heat and water would be beneficial for more vegetation production [89,91,92]. In this study, Tmax was positively associated with primary and mixed forest productivity, but it negatively influenced secondary and single-species forest productivity. More interestingly, canopy productivity had a faster recovery rate than stem/wood productivity

among the different forest types. These results implied that the individual pattern of stand aboveground productivity fluctuation was caused by a response in canopy productivity, and this was favorable for future biomass accumulation in wood production [92,93]. Therefore, subtropical mountain forests, especially primary, secondary, and mixed forests, have the potential to accumulate biomass after large-scale climate anomalies and will still be carbon sinks for the future [89]. Meanwhile, the community position and plant diversity differences among different forest types may also prevent the uniform response of subtropical mountain ANPP to regional climate anomalies and emphasize the importance of further research in subtropical forest ecosystems.

## 5. Conclusions

Aboveground productivity from natural forests and plantations (including the primary, secondary, mixed, and single-species stand types) indicated that subtropical mountain forests (less than 100 years) will continue to be carbon sinks [89]. Local and regional climate changes can have unique and lasting effects on aboveground productivity across different forests in subtropical China. Monthly streamflow influences on aboveground productivity were first explored and showed a stronger correlation with aboveground productivity than precipitation at the study site. As a predictor, streamflow during spring was stronger in estimating stand aboveground productivity than precipitation across all research plots. Extreme temperature (minimum and maximum temperature), especially during dry seasons (October to December) and wet seasons (July and August), as well as streamflow during spring, had a close relationship with forest aboveground productivity and its components.

Spearman correlation analysis indicated that single-species forest productivity was not significantly influenced by the MEI (multivariate ENSO index) and DD (number of dry days). However, further CCA indicated that the maximum temperature, streamflow, and precipitation were positively correlated with primary and mixed forest productivity, but negatively correlated with secondary and single-species forest productivity. These results confirmed that aboveground productivity dynamics were controlled by multiple complicating climatic factors in subtropical China. More remarkably, the productivity dynamics among different stand types might be regulated by vegetation community position and plant diversity at different scales and thus prevent the uniform response of subtropical mountain ANPP to regional climate anomalies. Therefore, further analysis and long-term research will be necessary in subtropical forest ecosystems.

Because local climatic variables, such as extreme temperature and streamflow, are significantly associated with large-scale climate anomalies, potential declines or increases in precipitation and temperature projected by regional climate circulation models for subtropical China could result in direct reductions or increases in forest aboveground productivity. The findings could also be useful for forecasting climate-induced variations in forest aboveground productivity as well as for the selection of tree species for planting in reforestation practices.

**Supplementary Materials:** The following are available online at http://www.mdpi.com/1999-4907/10/1/71/s1, Table S1: Spearman correlation coefficients between monthly climate variables and monthly productivity variables during the study period 1981–2015(420 months). Table S2: Results of one-way analysis of variance (ANOVA) on seasonal detrended interannual variations in climate variables: multivariate ENSO index (MEI), minimum temperature (Tmin), maximum temperature (Tmax), precipitation (PRE), streamflow (FLO) and number of dry days (DD); as well as productivity variables: canopy production (CP), wood production (WP) and aboveground net primary production (ANPP) recorded during the period 1981–2015 at primary (Pri), secondary (Sec), mixed (Mix) and single-species (Sin) forest located in subtropical China. Table S3: Results of one-way analysis of variance (ANOVA) on interannual detrended seasonal variations in climate variables: multivariate ENSO index (MEI), minimum temperature (Tmin), maximum temperature (Tmax), precipitation (PRE), streamflow (FLO) and number of dry days (DD); as well as productivity variables: canopy production (CP), wood production (WP) and aboveground net primary production (ANPP) recorded during the period 1981–2015 at primary (Pri), secondary (Sec), mixed (Mix) and single-species (Sin) forest located in subtropical China. Table S4: Results of linear mixed effects models on seasonally detrended time series investigating significant trends in climate variables: multivariate ENSO index (MEI), minimum temperature (Tmin), maximum temperature (Tmax), precipitation (PRE), streamflow (FLO) and number of dry days (DD); as well as productivity variables: canopy production (CP), wood production (WP) and aboveground net primary production (ANPP) recorded during the period 1981-2015 at primary (Pri), secondary (Sec), mixed (Mix) and single-species (Sin) forest located in subtropical

China. Table S5: Results of lagged cross-correlations between seasonal detrended climate and productivity variables indicating maximum correlation coefficients at different time lags (0–36 month) for correlations between climate variables: multivariate ENSO index (MEI), minimum temperature (Tmin), maximum temperature (Tmax), precipitation (PRE), streamflow (FLO) and number of dry days (DD); as well as productivity variables: canopy production (CP), wood production (WP) and aboveground net primary production (ANPP) recorded during the period 1981–2015 at primary, secondary, mixed and single-species forest located in subtropical China.

**Author Contributions:** Conceptualization, Q.L.; Data curation, S.M.; Formal analysis, H.Z.; Funding acquisition, Y.L.; Investigation, H.Z. and S.M.; Methodology, G.Z., J.Y. and S.S.; Project administration, Q.L.; Supervision, Y.L.; Writing—original draft, H.Z.; Writing—review & editing, S.S.

**Funding:** This research was funded by National Key Research and Development Program of China (No.2016YFC0502605), and Guizhou Provincial Gross Ecosystem Product (GEP) Accounting Pilot Projection.

**Acknowledgments:** The fieldwork was aided by the Administration Bureau of Jiulian Mountain National Nature Reserve, which granted us permission to conduct surveys and procure samples in the site. We also acknowledge the reviewers for their constructive comments to improve the manuscript.

**Conflicts of Interest:** The authors declare no conflict of interest.

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
