# Peer review of "Exploring the Sensitivity of Subtropical Stand Aboveground Productivity to Local and Regional Climate Signals in South China"

_forests, doi:10.3390/f10010071_

Round 1
Reviewer 1 Report
Based on the specific suggestions provided by the reviewers on an earlier version of the submitted manuscript the authors have greatly improved some of the methods and statistical analysis evaluating synchronicity among vegetation parameters and climatic controls.
However, due to the fact that parts of the text could still be improved I would recommend to clarify some of the specific suggestions (provided below) before acceptance of the revised version:
Line 25: cloudiness over the tropical Pacific Ocean, was significantly correlated with (remove comma)
Lines 103: probability of natural disasters, such as insect outbreaks, etc. (include comma)
Line 563-564: Based on the seasonal decomposition method, the climatic and forest productivity parameters were decomposed into a trend component, seasonal component and mutation component to verify the regularities in interannual and seasonal fluctuations. (should be "residual component")
Line 993: Figure 8.why not show the variation of productivity variables ANPP/WP/CP as well?
Line 1173: and reduced by 20.8% during the next 2009-2011 ENSO transition period. (should be recovered by 21%, right?)
Some references are cited incorrectly, e.g. see below capitalised AUTHORNAMES:
Line 1676: 82. MEIER, I.C.; LEUSCHNER, C. Belowground drought response of European beech: fine root biomass and 1676 carbon partitioning in 14 mature stands across a precipitation gradient. GLOBAL CHANGE BIOL 2008, 1677 14, 2081-2095.
Please carefully recheck several spelling mistakes and inconsistencies in sentence structure and semantics before submitting the final manuscript version!
Author Response
Response to Reviewer 1 Comments
Comments and Suggestions for Authors:
Based on the specific suggestions provided by the reviewers on an earlier version of the submitted manuscript the authors have greatly improved some of the methods and statistical analysis evaluating synchronicity among vegetation parameters and climatic controls.
However, due to the fact that parts of the text could still be improved I would recommend to clarify some of the specific suggestions (provided below) before acceptance of the revised version:
(1) Line 25: cloudiness over the tropical Pacific Ocean, was significantly correlated with (remove comma)
(2) Lines 103: probability of natural disasters, such as insect outbreaks, etc. (include comma)
(3) Line 563-564: Based on the seasonal decomposition method, the climatic and forest productivity parameters were decomposed into a trend component, seasonal component and mutation component to verify the regularities in interannual and seasonal fluctuations. (should be "residual component")
(4) Line 993: Figure 8.why not show the variation of productivity variables ANPP/WP/CP as well?
(5) Line 1173: and reduced by 20.8% during the next 2009-2011 ENSO transition period. (should be recovered by 21%, right?)
(6) Some references are cited incorrectly, e.g. see below capitalised AUTHORNAMES:
Line 1676: 82. MEIER, I.C.; LEUSCHNER, C. Belowground drought response of European beech: fine root biomass and 1676 carbon partitioning in 14 mature stands across a precipitation gradient. GLOBAL CHANGE BIOL 2008, 1677 14, 2081-2095.
(7) Please carefully recheck several spelling mistakes and inconsistencies in sentence structure and semantics before submitting the final manuscript version!
Response: Thank you for your careful work. We have made accordingly some changes in the revised manuscript as following:
(1) thank you for your advice. It has been corrected in the revised manuscript.
(2) thank you for your careful work. It has been corrected in the revised manuscript.
(3) thank you for your careful work. The problem has been corrected in the revised manuscript.
(4) it’s a good idea. In order to differ the ANPP, WP and CP from different forest types, so we use a single figure for each components of the four forest types and put 3 labels (ANPP, WP and CP) at the bottom of the figure.
(5) thank you very much for your careful work. The problem has been corrected in the revision. Due to a lower recovery rate (only 7.5%), it still needs further research.
(6) thank you for your careful work. The problem has been corrected in the revision. All references were checked carefully.
(7) we are very sorry for several spelling mistakes and inconsistencies in the manuscript. We have tried our best to make a correction in the revision.
Once again, thank you very much for your good comments. We really appreciate your advice in this study.
Reviewer 2 Report
The improved version is really much better.
Author Response
Response to Reviewer 2 Comments
Comments and Suggestions for Authors:
The improved version is really much better.
Response: Thank you very much for your good comments. We really appreciate your help in this study.
This manuscript is a resubmission of an earlier submission. The following is a list of the peer review reports and author responses from that submission.
Round 1
Reviewer 1 Report
Presented paper presents local unique findings which are in accordance with the general well known knowledge.
1. The description of method is little bit confusing, must be improved. Presentation of results are acceptable.
2. Regarding the discussion:
I have only one remarks- statement on autotrophic respiration - what about the heterotrophic one??
Reviewer 2 Report
General comments
This study investigating temporal variation in forest productivity of subtropical forest stands located in China represents an interesting analysis on the effects of projected increasing climate extremes, such as drought events on subtropical forests in China. Most interestingly, the study emphasizes the importance of local climate on the tree growth of different forests stands. However, despite the fact that the presented results are consistent with conventional findings that the growth of trees located at lower latitudes is less influenced by mean monthly temperature than seasonal patterns in precipitation, some of the presented methodology seems inappropriate to assess such temporal pattern based on the presentation of Pearson correlation coefficients between climate and productivity variables. Therefore, I would strongly recommend applying some more advanced statistical techniques instead of presenting correlation coefficients when analyzing synchronicity among vegetation parameters and climatic controls.
For instance, the finding that aboveground productivity for all plots was related to monthly precipitation of previous year and with annual precipitation, while almost all correlations did not reach the significant level on statistics (Figure 6) as well as that stand aboveground productivity for all plots showed significant negative correlation with monthly stream flow of previous and current year February and March (Figure 7), suggests that further analysis investigating the cross correlation between climate and productivity parameters should be conducted in order to asses respective effects of trend and seasonal signals.
As a result, the methods section presenting some of the statistical analysis investigating temporal variation and autocorrelation among vegetation productivity and climatic controls, and underlying statistical analysis should be further improved. For instance, there are several alternative statistical approaches, such as those based on decomposition of time-series, autoregressive models and cross-correlation functions, which allow assessing lag-time responses among respective variables. To that end, time series analysis accounts for respective trends of inter-annual, and seasonal fluctuations by decomposing respective components, which subsequently should be further investigated using lag-analysis (for instance see Hofhansl et al. 2014).
However, any time series may have a non-seasonal non-stationary and seasonal non-stationary component. Hence, to ensure a sound experimental design one has to ensure that all observations were independent, i.e. by verifying that the time series is stationary, which means that data have no upwards or downward trend. For a time series to be stationary, the mean, variance and co-variance of the time series at any given time interval should be constant. For this to happen it is necessary to account for seasonality and trend by decomposing the time series into trend/seasonal/reminder component to evaluate significant fluctuations and cross-correlation among respective signals (e.g. see description presented in HOFHANSL et al. 2014). First, time series analysis and seasonal decomposition of time series into seasonal, trend, and irregular components could be used to test for patterns of interannual and seasonal variations of climate and aboveground productivity. Second, linear mixed effect models (with interannual trend as fixed factor and seasonal variation as random effect) could be used to test for significant trends among these variables. Third, lagged correlation analysis could be used to investigate maximum correlation coefficients between seasonally detrended climate and component variables at different time lags resulting correlation coefficients indicating significant relationships among the investigated parameters. Based on these statistical techniques it could be shown that respective components were indeed significantly affected by the climatic signal, and that there are lagged responses in recovery rates after dry or wet events in the time period studied.
Hence, while it may be correct to conclude that it could be useful to forecast climate-induced variation in stand aboveground productivity as well as selection of planting tree species for reforestation practices based on the findings presented in this study it might be worth to consider the suggested alternative approach in order to further assess the relationship between climate and productivity parameters following the procedure indicated above. Please also consider including additional references discussing some of these statistical procedures in more detail (e.g. Zuur et al. 2003).
Additional references to be considered
Hofhansl, F., Kobler, J., Ofner, J., Drage, S., Pölz, E.-M. & Wanek, W. (2014). Sensitivity of tropical forest aboveground productivity to climate anomalies in SW Costa Rica. Global Biogeochem. Cycles, 28, 1437–1454.
Zuur, A.F., Fryer, R.J., Jolliffe, I.T., Dekker, R. & Beukema, J.J. (2003). Estimating common trends in multivariate time series using dynamic factor analysis. Environmetrics, 14, 665–685.
Specific comments
The manuscript lacks some linguistic capability, resulting in several spelling mistakes and inconsistencies in sentence structure and semantics (see comments below):
Figure 1: please check spelling of legend “Mererological Station”
Figure 2: shows annual stand aboveground productivity based on tree ring analysis.
Figure 3: only 2 correlations show significant relationship between stand aboveground productivity and monthly climate data.
Line 48: temperature increase is not 0.8 but 1°C already check LIT (IPCC 2017).
Line 53: “ there have been many compelling reasons to establish”
Line 59: “compared to single species forests”
Line 63 “under poor site conditions”
Line 68 “natural disasters”
Line 127: please check: “for estimating the four stand aboveground productivity”
Line 275: please explain the meaning of the sentence “higher temperatures can increase evapotranspiration and thus leaf stomata are apt to close for reducing water loss, leading to lower CO2 uptake and reduced carbon assimilation rates”.
Line 279: the finding “previous report showed a roughly 2-fold increase in respiration rates when 279 the air temperature increase by roughly 10°C“ is commonly known as “Q10” in the literature.
Line 308: please consider revising “As a predictor, streamflow during spring was stronger to estimate stand aboveground productivity than precipitation across all research plots.”